# PeerJ

# The relative importance of DNA methylation and *Dnmt2*-mediated epigenetic regulation on *Wolbachia* densities and cytoplasmic incompatibility

Daniel P. LePage[1], Kristin K. Jernigan[1,2] and Seth R. Bordenstein[1,3]

[1] Department of Biological Sciences, Vanderbilt University, Nashville, TN, USA
[2] Department of Cell and Developmental Biology, Vanderbilt University, Nashville, TN, USA
[3] Department of Pathology, Microbiology and Immunology, Vanderbilt University, Nashville, TN, USA

## ABSTRACT

*Wolbachia pipientis* is a worldwide bacterial parasite of arthropods that infects germline cells and manipulates host reproduction to increase the ratio of infected females, the transmitting sex of the bacteria. The most common reproductive manipulation, cytoplasmic incompatibility (CI), is expressed as embryonic death in crosses between infected males and uninfected females. Specifically, *Wolbachia* modify developing sperm in the testes by unknown means to cause a post-fertilization disruption of the sperm chromatin that incapacitates the first mitosis of the embryo. As these *Wolbachia*-induced changes are stable, reversible, and affect the host cell cycle machinery including DNA replication and chromosome segregation, we hypothesized that the host methylation pathway is targeted for modulation during cytoplasmic incompatibility because it accounts for all of these traits. Here we show that infection of the testes is associated with a 55% increase of host DNA methylation in *Drosophila melanogaster*, but methylation of the paternal genome does not correlate with penetrance of CI. Overexpression and knock out of the *Drosophila* DNA methyltransferase *Dnmt2* neither induces nor increases CI. Instead, overexpression decreases *Wolbachia* titers in host testes by approximately 17%, leading to a similar reduction in CI levels. Finally, strength of CI induced by several different strains of *Wolbachia* does not correlate with levels of DNA methylation in the host testes. We conclude that DNA methylation mediated by *Drosophila*'s only known methyltransferase is not required for the transgenerational sperm modification that causes CI.

Corresponding authors
Daniel P. LePage,
danielplepage@gmail.com
Seth R. Bordenstein,
s.bordenstein@vanderbilt.edu

## INTRODUCTION

*Wolbachia pipientis*, an obligate intracellular bacteria, is estimated to infect approximately 40% of all arthropod species (*Zug & Hammerstein, 2012*). This widespread prevalence can be attributed to efficient maternal transmission of the infection, intermediate rates of horizontal transmission to new hosts, and strong manipulations of the host reproductive

system to enhance its maternal transmission (*Stouthamer, Breeuwer & Hurst, 1999*; *Serbus et al., 2008*). These sexual alterations all act to increase the number of infected females within a population and include male-killing, feminization, parthenogenesis, and cytoplasmic incompatibility (CI). CI is the most common defect observed in *Wolbachia*-infected hosts and has been documented in numerous species (*Serbus et al., 2008*).

CI acts as a post-fertilization mating barrier by preventing the development of embryos from uninfected females that are mated with *Wolbachia*-infected males. This zygotic defect can be rescued, however, by females infected with the same strain of *Wolbachia* present in the male. This rescue capability gives a strong fitness advantage to *Wolbachia*-infected females and can lead to rapid sweeps of the infection through host populations. For instance, CI-inducing *Wolbachia* have been able to spread across most of the *Drosophila simulans* population in eastern Australia in less than a decade (*Kriesner et al., 2013*). CI is also a major isolation barrier between young sibling species (*Bordenstein, O'Hara & Werren, 2001*; *Jaenike et al., 2006*; *Miller, Ehrman & Schneider, 2010*) and is currently being used as a genetic drive mechanism to eliminate dengue virus in *Aedes aegypti* populations (*Moreira et al., 2009*; *Bian et al., 2010*; *Walker et al., 2011*) and to generally reduce mosquito population sizes (*Laven, 1967*; *O'Connor et al., 2012*).

The evolutionary, ecological, and medical importance of CI has fueled decades of research seeking to understand its underlying mechanisms. However, apart from studies that suggest the host genes JhI-26 and HIRA are involved (*Zheng et al., 2011*; *Liu et al., 2014*), it remains unknown how *Wolbachia* in the testes encode a sperm modification that renders embryos inviable. Previous work elucidated a few post-fertilization hallmarks of CI, most of which are associated with defects in the paternal genome during embryogenesis. These changes include a failure of maternal histones to deposit correctly, prolonged or incomplete replication of the paternal DNA, and failed condensation of the paternal chromosomes (*Breeuwer & Werren, 1990*; *Callaini, Dallai & Riparbelli, 1997*; *Landmann et al., 2009*). The alterations of the paternal chromatin and host cell cycle lead to a failure of the first mitosis followed by embryonic death. Interestingly, *Wolbachia* are not actually present within the sperm of their hosts, indicating a semi-permanent modification of the paternal genome that is transgenerationally transmitted to the egg (*Clark et al., 2008*).

Several assumptions can be made about the paternal genome modification underlying cytoplasmic incompatibility including:

(i) It targets host pathways that are highly conserved across numerous host species.
(ii) It involves a semi-permanent but reversible alteration to the paternal genome.
(iii) It must be able to affect histone recruitment, DNA replication, and chromosome condensation.

Working under these assumptions, we selected the host DNA methylation pathway as a probable target for *Wolbachia*. Methylation is a stable, yet reversible, modification to DNA that could be sex-specific and easily rescued by infected females. It also has the capability to modulate many cell cycle functions including chromosome condensation and histone recruitment (*Bird, 2001*; *Harris & Braig, 2003*; *Weber & Schübeler, 2007*) and has previously

been hypothesized to play a role in CI (*Negri, 2011*; *Saridaki et al., 2011*; *Ye et al., 2013b*; *Liu et al., 2014*). While the role of DNA methylation in insects is not fully understood, it is a highly conserved pathway that shows strong upregulation during embryogenesis (*Field et al., 2004*). Finally, the ability of bacteria to alter host methylation and chromatin structure is increasingly recognized (*Gómez-Díaz et al., 2012*; *Bierne, Hamon & Cossart, 2012*) and previous work shows that *Wolbachia* infection in particular alters the host methylation profile in both leafhoppers and mosquitoes (*Negri et al., 2009*; *Ye et al., 2013a*).

Here we use the model organism *Drosophila melanogaster* infected with the *w*Mel strain of *Wolbachia* to determine the role of host DNA methylation in CI. *D. melanogaster* flies utilize just one canonical DNA methyltransferase, *Dnmt2* (*Lyko, Ramsahoye & Jaenisch, 2000*), which enables easy genetic manipulation of the host methylation pathway without the confounding influence of other DNA methyltransferases (*Dnmt1* and *Dnmt3*) present in most other insect species (*Werren et al., 2010*). While the role of *Dnmt2*-dependent methylation is debated and multifaceted (*Schaefer & Lyko, 2010*; *Raddatz et al., 2013*; *Takayama et al., 2014*), evidence demonstrates that the methylation machinery in *D. melanogaster* is not only present but also functional (*Lyko, Ramsahoye & Jaenisch, 2000*; *Kunert et al., 2003*; *Schaefer, Steringer & Lyko, 2008*; *Gou et al., 2010*). Moreover, overexpression of the mouse *Dnmt3a* in *D. melanogaster* induces CI-like defects such as reduced rates of cell cycle progression and altered chromosome condensation (*Weissmann et al., 2003*).

## MATERIALS AND METHODS

### Fly rearing and dissections

All flies were reared on a cornmeal and molasses-based media at 25 °C. The *Dnmt2* loss-of-function mutant has been previously described (*Goll et al., 2006*). Briefly, the mutant contains a 28bp insertion with multiple stop codons as well as a frameshift within the coding region of *Dnmt2*. Overexpressing flies were created through the Gal4-UAS system. Crosses were performed between virgin *nos*-Gal4 driver females ($y^1w^*$; $P\{w[+mC] = \text{GAL4-nos.NGT}\}40$ (either *Wolbachia*-infected or uninfected)) and 5–6 uninfected UAS-*Dnmt2* (*Kunert et al., 2003*), UAS-GFP or $W^{1118}$ males. Crosses for Fig. S1 were conducted between virgin Act5c-Gal4 driver females ($y^1w^*$; $P\{w[+mC] = \text{Act5C-GAL4}\}25\text{FO1/CyO}, y^+$ , *Wolbachia* infected or uninfected depending on desired progeny) and UAS-*Dnmt2* males. For Act5c-Gal4 crosses, straight-winged progeny were assumed to be overexpressing *Dnmt2* while CyO expressing lines were used as the wild-type expressing lines. *Wolbachia*-uninfected lines were created through tetracycline treatment (20 ug/mL for 3 generations) and infection status was confirmed through PCR using the following primers: WolbF (GAAGATAATGACGGTACTCAC) and WolbR3 (GTCACT-GATCCCACTTTAAATAAC) which target the 16S rRNA gene of *Wolbachia*. These lines were further reared for at least three generations on undrugged media before experimentation to avoid detrimental paternal effects seen in other systems (*Zeh et al., 2012*).

The *w*Au, *w*No, and *w*Ri (also known as *w*Ri Agadir) strains of *Drosophila simulans* were kindly provided by Charlat Sylvain (University of Lyon, France). All testes and ovary dissections were performed in cold phosphate buffered saline (PBS). Males were dissected

within 24 h of emergence while females were aged 3–4 days before dissections. Testes samples consisted of tissue obtained from a minimum of 20 males while ovary samples were pooled from 10 females each. Tissues were frozen and stored at $-80\,°C$ before analysis.

### Hatch rate assays

Assays were performed using a grape juice/agar media in 30 mm plates for egg laying. For each cross 32–48 individual crosses of one male and one female were set up in separate mating chambers with individual grape juice plates. A minimal amount of a 1:2 dry yeast and water mix was added to each plate and the parents were allowed to mate for 16 h before the grape juice plates were discarded. Fresh plates were then used for 24 h, removed, and the number of eggs laid was counted for each cross. The number of unhatched eggs was counted again at 36 h after the plates had been removed to determine hatch rates.

### MethylFlash quantification of DNA methylation

Genomic levels of cytosine methylation (5-mC) were measured using the MethylFlash kit (Epigentek, Farmingdale, NY, USA). 8–10 replicate sets of testes (20–40 testes pairs each replicate) were dissected and DNA was isolated using the Puregene Tissue kit (Qiagen, Venlo, Netherlands). 100 ng of genomic DNA from each sample was used and each sample was analyzed in duplicate on a BMG LabTech FLOUstar OPTIMA plate reader (Ortenberg, Germany) according to manufacturer instructions.

### Wolbachia density

Eight replicates each of whole animals (pools of 3), testes (pools of 20 pairs), and ovaries (pools of 10 pairs) were collected and DNA was isolated. All males were less than 24 h old while females had been aged 3–4 days. Quantitative PCR was performed on a Bio-Rad CFX96 Real-Time System using iTaq Universal SYBR Green Supermix (Bio-Rad, Hercules, CA, USA). *groEL* copy number, determined against a standard curve, was compared to counts for the host gene *Actin*, also determined against a standard curve. It was assumed that one copy of *groEL* was present in each *Wolbachia* genome and 1 or 2 copies of Act5c (for males and females, respectively, as the gene is on the X chromosome) in each *Drosophila* genome. Primers: Act5c (231bp product, Forward: ATGTGTGACGAA­GAAGTTGCT Reverse: GTCCCGTTGGTCACGATACC), *groEL* (97bp product, Forward: CTAAAGTGCTTAATGCTTCACCTTC Reverse: CAACCTTTACTTCCTATTCTTG). qPCR conditions: 50° 10 min, 95° 5 min, 40× (95° 10 s, 55° 30 s), 95° 30 s. Followed by melt curve analysis (0.5° steps from 65–95° for 5 s each).

### Gene expression

Quantitative PCR was performed on a Bio-Rad CFX96 Real-Time System using iTaq Universal SYBR Green Supermix (Bio-Rad, Hercules, CA, USA). RNA was isolated from 8 sets of testes (20 pairs each) using the RNeasy Mini kit (Qiagen, Venlo, Netherlands) and DNA was removed with the TURBO DNA-free DNase kit (Ambion, Grand Island, NY, USA). cDNA was synthesized using a SuperScript III First-Strand kit (Invitrogen, Grand Island, NY, USA) and diluted 1:20. All calculations were done using delta delta Ct

with Rp49 expression used for normalization of results. Primers: *Dnmt2* (150bp product, Forward: CCGTGGCGTGAAATAGCG Reverse: ACACCGCTTTCGGAGGACG), Rp49 (154bp product, Forward: CGGTTACGGATCGAACAAGC Reverse: CTTGCGCTTCTTG-GAGGAGA). qRT-PCR conditions are the same as used in qPCR for *Wolbachia* densities.

### Bisulfite sequencing

One hundred testes were dissected in PBS from *Wolbachia* infected ($y^1w^*$) and uninfected males and flash frozen. gDNA was then isolated using the Puregene kit (Qiagen, Venlo, Netherlands) and fragmented by Covaris shearing. gDNA was submitted to Vanderbilt Technologies for Advanced genomics (VANTAGE) where the PE-75 bp library was generated using the TruSeq sample preparation kit (with methylated adapters), bisulfite treated, PCR amplified (EpiMark and ZymoTaq) and sequenced (Illumina HiSeq 2000, 86bp PE read). Sequences with $\geq 10\times$ coverage were analyzed using Bismark (*Krueger & Andrews, 2011*) and cytosines which were methylated in at least one read were counted.

## RESULTS AND DISCUSSION

### *Wolbachia w*Mel increases levels of testes DNA methylation

MethylFlash analysis of host DNA from testes revealed that infection with the *Wolbachia* strain *w*Mel in *Drosophila melanogaster* increases levels of genome-wide cytosine methylation (Fig. 1A). More importantly, this methylation is specific to the host testes (55% increase, $P = 0.0015$, Mann Whitney U test) and is not observed in the ovaries, consistent with the prediction that only the paternal genome is modified during cytoplasmic incompatibility. The overall levels of methylation are extremely low, which is consistent with previously reported levels of methylation in *Drosophila melanogaster* (*Lyko, Ramsahoye & Jaenisch, 2000*; *Kunert et al., 2003*). Conflicting reports over the strength and prevalence of DNA methylation in *D. melanogaster* (*Lyko, Ramsahoye & Jaenisch, 2000*; *Raddatz et al., 2013*; *Schaefer & Lyko, 2010*) led us to test the validity of our initial results with genome-wide bisulfite sequencing. Results indicate that, contrary to most other species, DNA methylation in *Drosophila melanogaster* is not CpG specific and is evenly distributed over cytosine residues (Fig. 1B and Table S1). Sequencing results also mirror those of MethylFlash and show that infection with *w*Mel increases testes DNA methylation 46% across all cytosine residues with a range of 43–54% depending upon the type of cytosine residue (CpG, CHG, or CHH) (Fig. 1B and Table S1). The minor discrepancies between MethylFlash and bisulfite sequencing (55% and 46% increase in methylation, respectively) are likely due to the sensitivity of the MethylFlash system on such low quantities of methylation. A more thorough investigation of the bisulfite sequencing, including changes in promoter and gene body methylation, is ongoing.

### Overexpression of *Dnmt2* neither induces nor strengthens CI

*Drosophila melanogaster* possess just one canonical DNA methyltransferase, *Dnmt2*, and overexpression of this enzyme in fruit flies has previously been shown to increase levels of DNA methylation (*Kunert et al., 2003*; *Schaefer, Steringer & Lyko, 2008*). Utilizing the Gal4-UAS expression system, we overexpressed *Dnmt2* in uninfected males to test

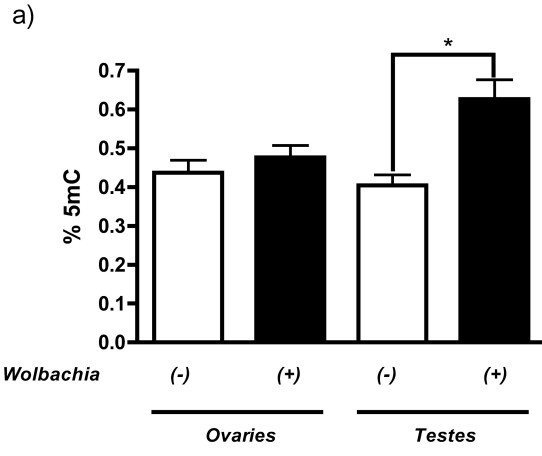

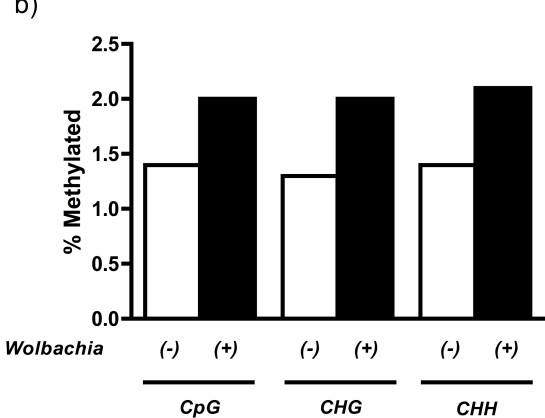

**Figure 1 *Wolbachia* increase host levels of DNA methylation.** (a) *Wolbachia* infection (*w*Mel) of *Drosophila melanogaster* increases DNA methylation in host testes by 55% ($P = 0.0015$, Mann–Whitney U (MWU) test, two-tailed), as measured by the ELISA-based MethylFlash kit. This increase is not observed in host ovaries ($P = 0.25$). Bars denote standard error of the mean (SEM) (b) Bisulfite sequencing of *Drosophila melanogaster* testes DNA shows that infection by *w*Mel increases methylation of all cytosine residues including CpG (43%), CHG (54%), and CHH (50%).

if an increase in host methylation alone could induce the CI defect of reduced embryo hatching rates. Figure 2 shows that there was no discernable difference in hatching rates with uninfected males expressing increased or wild type levels of *Dnmt2* ($P = 0.91$, MWU). The result was confirmed using an Actin-based driver that again yielded no discernable differences in hatch rates compared to wild type flies (Fig. S1, $P = 0.83$, MWU). These findings specify that amplified levels of *Dnmt2*-mediated epigenetic regulation are not sufficient to recapitulate cytoplasmic incompatibility.

If multiple factors are responsible for CI, it is possible that overexpression of *Dnmt2*, while unable to induce CI in uninfected flies, may be able to strengthen the modification in the presence of *Wolbachia*. To test this hypothesis, we overexpressed *Dnmt2* in *Wolbachia*-infected males that were then mated to uninfected virgin females. Surprisingly, *Dnmt2* overexpression in males decreased the level of cytoplasmic incompatibility by an average of 17.4% (Fig. 3). This effect is not dependent on the *Dnmt2* expression status

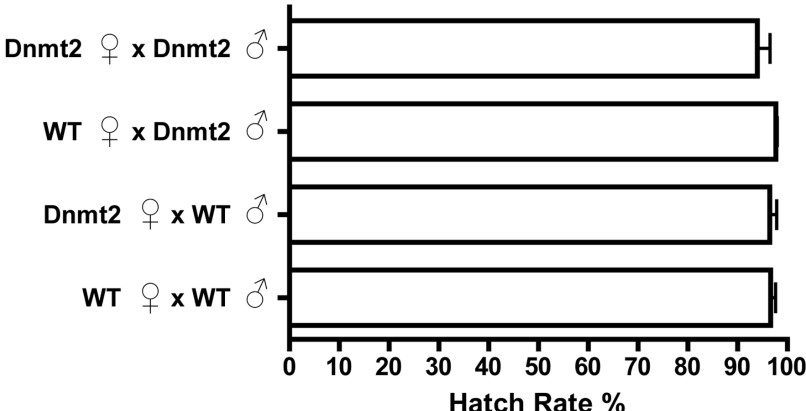

**Figure 2 Expression of DNA methyltransferase 2 does not induce CI.** Overexpression of the DNA methyltransferase *Dnmt2* in uninfected males, utilizing the Gal4-UAS system with a *nos* driver, does not reduce hatching rates. *Dnmt2*, overexpressing flies; WT, wild type flies. Bars denote standard error of the mean (SEM).

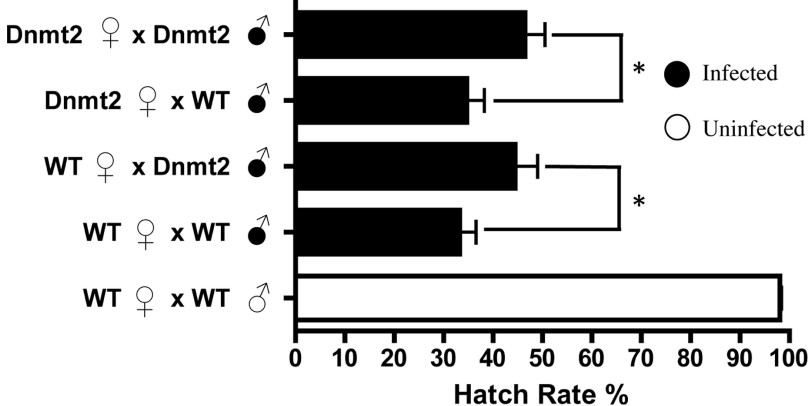

**Figure 3 Overexpression of *Dnmt*2 reduces levels of CI.** The overexpression of *Dnmt2* in *Wolbachia*-infected males decreases rates of CI ($P < 0.05$, Mann–Whitney U test). *Dnmt2* expression in the mother has no effect. Bars denote standard error of the mean (SEM). *Dnmt2*, overexpressing flies; WT, wild type flies.

of the female and suggests that increased methylation of host DNA can diminish the penetrance of cytoplasmic incompatibility.

## Overexpression of *Dnmt*2 reduces *Wolbachia* titers in host testes

Previous work suggested that *Dnmt2* is detrimental to *Wolbachia* proliferation in mosquitoes. In fact, *Wolbachia* strain *w*Mel Pop-CLA utilizes a host miRNA to downregulate *Dnmt2* expression when infecting *Aedes aegypti* (*Zhang et al., 2013*). We observed no differences in Dnmt2 expression between *w*Mel infected and uninfected *D. melanogaster* testes (data not shown) but hypothesized that overexpression of *Dnmt2* in the host may adversely affect *Wolbachia* titers. In support of this prediction, we found that *Wolbachia* density (as measured by the ratio of *Wolbachia groEL* gene copy number/*Drosophila Actin* gene copy number) decreased by 17.3% in adult testes overexpressing *Dnmt2* transcripts

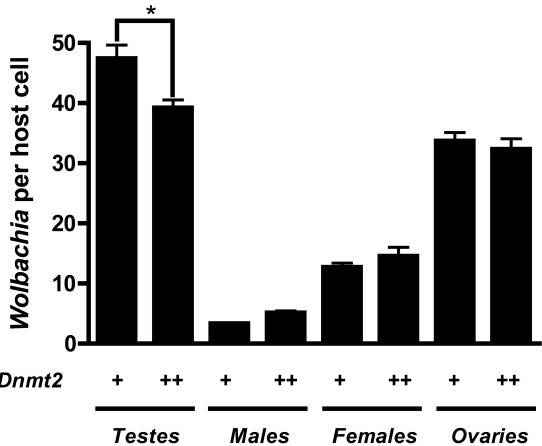

**Figure 4** *Dnmt2* **overexpression alters** *Wolbachia* **titers.** Overexpression of *Dnmt2* reduces *Wolbachia* titers within the testes (*P* < 0.01, MWU test) but has no affect on titers within ovaries or whole flies. *Wolbachia* infection is derived from the $y^1w^*$ *Drosophila* background. Bars denote standard error of the mean (SEM). *Dnmt2* ++, overexpressing flies; *Dnmt2* +, wild type flies. (*P* = 0.007, Mann–Whitney *U* test, two-tailed).

by 9.6% (Fig. 4 and Fig. S2, respectively). The low level of transcript overexpression could be specific to the developmental stage of the experimental sample or due to usage of a pUAST vector for germline expression instead of the more efficient pUASP (*Kunert et al., 2003*). While the upregulation of *Dnmt2* in infected males is not statistically significant, it remains possible that actual protein levels are much higher than those represented by RNA transcripts. Strong protein expression, as measured by Western blot, was seen in uninfected ovaries (data not shown). Nevertheless, the 17.3% decrease in *Wolbachia* titers compares well with the 17.4% reduction in CI penetrance reported above. Expression of the negative control green fluorescent protein (GFP) did not reduce *Wolbachia* titers, as expected (Fig. S3). As the bacterial and phage density models of CI specify that *Wolbachia* titers in the testes are linked to the strength of CI (*Breeuwer & Werren, 1993*; *Bordenstein et al., 2006*), we conclude that the reduction of CI observed in *Dnmt2*-overexpressing males is likely due to reduced *Wolbachia* density.

Even though we do not observe any change in *Dnmt2* mRNA levels after *Wolbachia* infection, we cannot rule out that *Wolbachia* may be affecting intracellular *Dnmt2* localization rather than levels of gene expression. An increase in localization of *Dnmt2* to the nucleus would not only protect the cytosolic *Wolbachia* but also explain the additional genomic methylation associated with infection. In this scenario, the testes-specific increase in host methylation initially observed would simply be a by-product of high *Wolbachia* activity. Additionally, an immunomodulatory role for *Dnmt2* in *Drosophila* has already been documented in protection against RNA viruses (*Durdevic et al., 2013*) though we believe the findings in this report are the first evidence for a putative antibacterial role for *Dnmt2* in fruit flies.

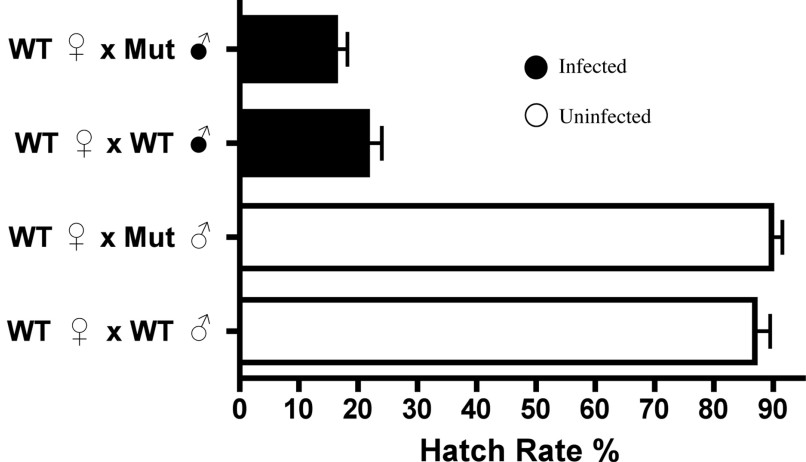

**Figure 5 *Dnmt2* mutants express wild-type levels of CI.** Crosses with *Dnmt2*-mutant males ("Mut") show that *Dnmt2* expression within the father is not necessary for expression of CI. Mut, *Dnmt2* mutant flies; WT, wild type flies. Bars denote standard error of the mean (SEM).

## Hosts defective in DNA methylation still exhibit CI

As *Dnmt2* overexpression did not induce nor increase cytoplasmic incompatibility, we next tested the strength of CI in hosts defective in the methyltransferase pathway. Knockout mutants for *Dnmt2* characterized by *Goll et al. (2006)* were acquired and found by PCR and amplicon sequencing to be infected by the *w*Mel strain of *Wolbachia*. The strain is hereafter referred to as Mut and was tetracycline treated for three generations to create the uninfected line MutT. We show by MethylFlash that the increase in host DNA methylation induced by *Wolbachia* infection is abolished in the knockout Mut background (Fig. S4) and is thus *Dnmt2*-dependent. However, loss of this crucial enzyme in the DNA methylation pathway has no effect on the penetrance of CI (Fig. 5), as shown in comparisons between mutant and wild type males mated to uninfected females ($P = 0.13$, MWU). The low level of DNA methylation still present in mutants has recently been observed by others (*Boffelli, Takayama & Martin, 2014*) and suggests a possible mechanism of DNA methylation in *Drosophila* that is independent of canonical DNA methyltransferases. Thus, it is possible that CI could be induced by alterations in genomic methylation but in a *Dnmt2*-independent manner.

Curiously, despite the previously observed role for *Dnmt2* in host immunity (*Zhang et al., 2013*; *Durdevic et al., 2013*), the mutants observed here exhibit no increase in *Wolbachia* titers within any of the tissues tested (Fig. S5). It is interesting to note that *Dnmt2* mutant *Drosophila*, derived from the W[1118] background line, have titers that are, on average, half of those seen in *y[1]w\** background lines (see Fig. 4). This difference has been observed several times in our experiments and suggests either a differing ability of the host lines to control *Wolbachia* titers or an as yet unclassified difference in the *w*Mel strains infecting these flies.

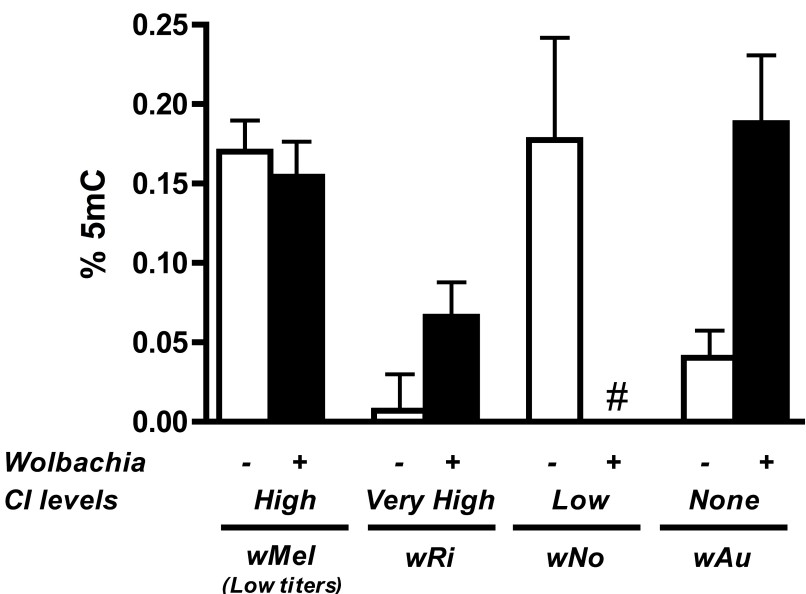

**Figure 6 Levels of host DNA methylation do not correlate with strength of CI.** Testing of several different *Wolbachia* infections, capable of inducing various levels of CI in their respective hosts, shows that levels of host DNA methylation and strength of CI are not correlated. Bars denote standard error of the mean (SEM) of testes DNA methylation, as measured by MethylFlash. White bars (−) denote uninfected flies and black bars (+) denote infected flies. # indicates levels of methylation too low for detection.

## Host levels of DNA methylation do not correlate with strength of CI

To substantiate the claim that DNA methylation is not involved in the induction of CI for other *Wolbachia* strains and/or host species, we tested the DNA methylation status of testes DNA from *Drosophila* species infected with various strains of *Wolbachia*. These taxa include *D. simulans* infected with strains *w*Ri, *w*No, and *w*Au, which express strong, moderate, and no CI, respectively. We also tested a different *D. melanogaster*-infecting strain *w*Mel derived from the $W^{1118}$ background strain instead of $y^1w^*$. As previously mentioned, while the $W^{1118}$ line induces strong CI, *Wolbachia* titers are much lower in these animals compared to the infection found in $y^1w^*$.

Results show that methylation status of the infected host testes is random with regards to the strength of CI (Fig. 6). While infection with the high CI-inducer *w*Ri exhibits higher methylation in infected testes as compared to uninfecteds, this effect is marginally insignificant ($P = 0.072$, MWU) and is countered by data from the *w*Au strain, which causes no CI but still significantly increases host DNA methylation in testes ($P = 0.0047$, MWU). Furthermore, infection with the *w*No strain of *Wolbachia*, which causes moderate CI, actually has less methylation in host testes. Finally, a low-titer infection of *w*Mel ($W^{1118}$), while still inducing CI, does not induce the same level of DNA methylation associated with a high-density infection ($y^1w^*$).

## CONCLUSIONS

The underlying mechanism of *Wolbachia*-induced CI largely remains elusive after several decades of research. Here we show that host DNA methylation, a promising candidate pathway hypothesized to play a role (*Negri, 2011*; *Saridaki et al., 2011*; *Ye et al., 2013b*; *Liu et al., 2014*), does not seem to be involved in the induction of CI. While *Wolbachia* infection preferentially increases host DNA methylation in *Drosophila melanogaster* testes (Fig. 1), this modification is not conserved across other CI-causing strains of *Wolbachia* (Fig. 6) and overexpression of a host methyltransferase neither induces nor increases rates of CI. We have also found that *Wolbachia*-induced changes in host methylation are dependent on the DNA methyltransferase *Dnmt2* (Fig. S4) but that *Drosophila melanogaster* lacking *Dnmt2* still suffer from CI (Fig. 5). Finally, we found *Dnmt2* has anti-*Wolbachia* properties, as previously reported in *Aedes aegypti* (*Zhang et al., 2013*), and overexpression of *Dnmt2* reduces the strength of CI.

Taken together, we show that one of the canonical chromatin modification pathways, *Dnmt2*-dependent DNA methylation, likely has no role in *Wolbachia*-induced CI. *Wolbachia* infection can be associated with changes in host methylation levels, but it is most likely a consequence of the bacteria modulating host immune response or the host defending itself against the infection. The possibility also remains that infection alters gene-specific, and *Dnmt2*-independent, levels of methylation that our current study of genomic methylation levels has not detected. While further investigation of the *Dnmt2* epigenetic pathway will not elucidate a CI mechanism, it may be useful in studying the complex nature of pathogen-host interactions between *Wolbachia* and the many species it infects. It remains possible that a novel methyltransferase, recently suggested to exist in *Drosophila* (*Takayama et al., 2014*; *Boffelli, Takayama & Martin, 2014*), could affect CI.

## ACKNOWLEDGEMENTS

We are grateful to Rini Pauly for computational assistance, and to the four reviewers and the editor for constructive feedback.

### Funding

This research was made possible by NIH awards R01 GM085163 and NSF DEB 1046149 to SRB. The funders had no role in study design, data collection and analysis, decision to publish, or preparation of the manuscript.

### Grant Disclosures

The following grant information was disclosed by the authors:
NIH: R01 GM085163.
NSF DEB: 1046149.

### Competing Interests

The authors declare there are no competing interests.

## Author Contributions

- Daniel P. LePage conceived and designed the experiments, performed the experiments, analyzed the data, wrote the paper, prepared figures and/or tables, reviewed drafts of the paper.
- Kristin K. Jernigan conceived and designed the experiments, performed the experiments, analyzed the data, prepared figures and/or tables, reviewed drafts of the paper.
- Seth R. Bordenstein conceived and designed the experiments, contributed reagents/materials/analysis tools, wrote the paper, prepared figures and/or tables, reviewed drafts of the paper.

## DNA Deposition

The following information was supplied regarding the deposition of DNA sequences:

The data discussed in this publication have been deposited in NCBI's Gene Expression Omnibus (*Edgar, Domrachey & Lash, 2002*) and are accessible through GEO Series accession number GSE63795.

## Supplemental Information

Supplemental information for this article can be found online at http://dx.doi.org/10.7717/peerj.678#supplemental-information.

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
