# Peer review of "The relative importance of DNA methylation and Dnmt2-mediated epigenetic regulation on Wolbachia densities and cytoplasmic incompatibility"

_PeerJ, doi:10.7717/peerj.678_

## Round 0.1 · original submission · Minor Revisions

As you can see, the reviewers unanimously thought the results and overall study interesting. However, several pointed out the trim materials and methods section -- please expand this such that someone could replicate the results (especially with regards to the fly stocks and the bioinformatics). Also, reviewer #3's concerns about the strength of the conclusions and alternative interpretations of the data resonated with me -- please address these concerns in the discussion.

Reviewer 1 ·

Basic reporting

The manuscript by LePage et al looks into the possibility of differential DNA methylation in Drosophila and its effect on cytoplasmic incompatibility (CI), an effect whose underlying molecular mechanism remains unresolved. Since there is only one known DNA methyltransferase (Dnmt2) in the Drosophila genome, this makes studying the effects relatively easier. The main results in this study are: 1) Wolbachia infection leads to increased Dnmt2 levels leading to increased genome methylation, 2) overexpression of Dnmt2 in non-infected flies has no effect on CI, 3) overexpression of Dnmt2 in Wolbachia-infected flies leads to reduced Wolbachia density and hence reduced CI effect, 4) Dnmt2 mutation had no effect on CI, and 5) no correlation was found between the levels of genome DNA methylation and CI intensity effects in different hosts and Wolbachia strains. Overall, the results clearly show that genome methylation altered by Wolbachia does not appear to be responsible for CI.

Experimental design

This is a well-executed research, technically sound.

Validity of the findings

Results justify the conclusions.

Additional comments

I only have very minor comments.

1) There is one study which shows a link between expression of hira and its effect on CI. It is worth citing and using it in the discussion.
Ref: Zheng Y, Ren P-P, Wang J-L, Wang Y-F (2011) Wolbachia-Induced Cytoplasmic Incompatibility Is Associated with Decreased Hira Expression in Male Drosophila. PLoS ONE 6(4): e19512

2) Line 233: affects ==> effects

3) Volume/page/article number (for e-journals) are missing from the following references:
Bian et al
Durdevic et al
Gou et al
Jaenike et al
Liu et al
Raddatz et al
Saridaki et al
Schaefer et al
Zhang et al

·

Basic reporting

1. I suggest to change the title to be more conclusive.
2. The material and method section looks as if it was meant to be at the end of the manuscript (see below).
3. The result and discussion section is fluent, however, figure legends should contain more information.
Fig 4. Name the lines background to be consistent with the text. I'm not convinced the over-expressing lines have half Wolbachia density. What are the units for Y?
Fig S3 and S4 are not in the write order according to their legends.
4. Discussion may refer to alternative explanations such as JH and Mst84Db.
5. The conclusion is overlapping the discussion.I believe this section should be a brief summary and not include more intensive discussion, thus I think L237-252 should be implanted the discussion.

Experimental design

The material and method section should be written more informatively and explain the experimental design clearly:
First section is very brief, more info is need on the experimental design: name the fly strains as appear in the following text and explain how they were obtained. What crosses were made and for what purpose?
Please explain abbreviations when first appear.
Please mention how old where the flies when dissected or frozen.
Please add manufactures' location.
L81 vs. L93, minimum of 20 male?
Please add a brief sentence explaining what actually the kit measures (i.e. what is 5-mC).

Validity of the findings

1. L146. 55% vs. 46% methylation between the two methods, is this an acceptable difference? Is it significant?
2. Were the overexpressing lines (both not infected and infected) actually tested for over expression? It is only mention for the later for the infected lines, and it is not clear if the 9.6% increase is significant. If it is not, than it cannot explain the observed change in Wolbachia density and CI.

Additional comments

This is an intersting paper on the way to understand the mechanism behind CI

Reviewer 3 ·

Basic reporting

No comments

Experimental design

Methods presentation needs improvement – see below

Validity of the findings

Raw data are currently not available – see below

Additional comments

LePage et al. present a study on the influence of DNA methylation on Wolbachia-induced CI. The study is clearly written and a valuable contribution to the understanding of CI. I have three major criticisms; these solved, the paper can be cleared for acceptance.

(1) The Mat&Met section is very brief, and the information provided is not always sufficient to reproduce the experiments. I suggest to not change the current main text (as it gives a nice overview), but provide a technical supplement with all lab procedures in detail.

(2) The authors finally conclude (l 253 f) that "DNA methylation, has no role in Wolbachia induced cytoplasmic incompatibility". I fear that this conclusion is a bit overhasty, as it is mainly based on the results from global methylation proportions in reproductive tissues and overexpression/knockout of Dnmt2. Two scenarios can be easily imagined that would methylation still allow playing a role in CI: (i) an alternate methylation pathway besides Dnmt2 (briefly mentioned by the authors in l 259) could be the relevant CI locus, and (ii), targeted methylation of one or a few important genes by Wolbachia would only minimally alter the global methylation rate. Both scenarios should be discussed.

(3) In frame of this study, a methylation-sensitive bisulfate Illumina approach was conducted. The Met&Met are too vague to really understand the NGS design (How many reads? PE insert size[s]? Overall sequencing quality? Bioinformatic pipeline? etc.), but I was disappointed to see that this effort resulted in only one panel in Fig. 1, again presenting only global data. The authors should sit on a wealth of NGS data suitable to elucidate much more how Wolbachia is interacting with the Drosophila methylation machinery. I would like to see more in depth bioinformatic analyses of these data, and of course the raw data uploaded to NCBI SRA. If the authors do not want to present these results here (maybe they will be the core for another paper), the whole NGS passage and Fig 1b should be removed.

Minor issue: In Fig. 6, the meaning of + and – marks should be explained in the caption.

·

Basic reporting

Experimental details could be clearer. Example: What is the age (developmental stage) of the testes/ovaries used in Figure 1. In general, I thought that I could make reasonable inferences and that the details were unlikely to have major effects on the results, but if I were asked the critical reviewing questions: "Would I expect to be able to repeat the experiments?", the answer would be "no".

Experimental design

None.

Validity of the findings

No reservations.

Additional comments

The article is clear and straightforward and I think that the conclusions are probably correct. It's not relevant for this journal, but I think that they are also important, because they throw real doubt on an explanatory inference that has been around for awhile. I ask only that you re-read to add details where you can, so that it would be possible to reproduce the experiments. I provided one example above.

---

## Round 0.2 · accepted · Accept

All reviewers are well satisfied by the changes you've implemented in this revised manuscript.

Reviewer 1 ·

Basic reporting

The authors have addressed the comments I made.

Experimental design

NA

Validity of the findings

NA

Additional comments

NA

·

Basic reporting

no comments

Experimental design

no comments

Validity of the findings

no comments

Additional comments

All my concerns where answered, methodology and writing are much clearer now. If the authors prefer to leave the title as is, OK.

Reviewer 3 ·

Basic reporting

No comments/criricism

Experimental design

No comments/criricism

Validity of the findings

No comments/criricism

Additional comments

The authors have thoroughly revised the manuscript and properly addressed all criticism. IMHO, the paper can now be accepted as is.